# Metabolic gene alterations impact the clinical aggressiveness and drug responses of 32 human cancers

Musalula Sinkala [1]*, Nicola Mulder[1] & Darren Patrick Martin[1]

Malignant cells reconfigure their metabolism to support oncogenic processes such as accelerated growth and proliferation. The mechanisms by which this occurs likely involve alterations to genes that encode metabolic enzymes. Here, using genomics data for 10,528 tumours of 32 different cancer types, we characterise the alterations of genes involved in various metabolic pathways. We find that mutations and copy number variations of metabolic genes are pervasive across all human cancers. Based on the frequencies of metabolic gene alterations, we further find that there are two distinct cancer supertypes that tend to be associated with different clinical outcomes. By utilising the known dose-response profiles of 825 cancer cell lines, we infer that cancers belonging to these supertypes are likely to respond differently to various anticancer drugs. Collectively our analyses define the foundational metabolic features of different cancer supertypes and subtypes upon which discriminatory strategies for treating particular tumours could be constructed.

---

[1] Computational Biology Division, Department of Integrative Biomedical Sciences, University of Cape Town School of Health Sciences, Anzio Rd, Observatory, Cape Town 7925, South Africa. *email: smsinks@icloud.com

The transformation of normal cells into cancer cells requires the adaptation of multiple metabolic processes to satisfy the high energy demands of malignant cellular growth, proliferation and survival[1,2]. Accordingly, metabolic dysregulation is recognised as a hallmark of malignant cellular phenotypes[3,4]. Although many of the metabolic processes occurring in cancer cells are similar to those occurring in healthy proliferating cells, a series of genetic and epigenetic modifications in cancer cells can result in the aberrant regulation of these processes[5,6]. Among these genetic alterations are those occurring in a range of genes that are involved in metabolism. These alterations include diverse driver mutations and gene copy number alterations, which can impart a substantial degree of metabolic heterogeneity to different tumours of the same cancer type[7]. There is, therefore, keen interest in determining how genetic alterations within various types of malignant cells relate to specific aspects of the metabolic dysregulation occurring within these cells.

Transcriptomic and metabolomic analyses of various human tumours have revealed the numerous metabolic peculiarities of cancer cells that likely play essential roles in oncogenesis and cancer progression[7-12]. In general, these peculiarities can be traced to abnormal variations in the expression levels of either particular metabolic enzymes or the proteins that regulate these enzymes[13,14]. These and other studies[1,3,6,15-19] have also yielded a growing appreciation of how the aberrant metabolic changes in cancer cells influence the anticancer drug responses of different tumours.

Besides enabling the selection of the most appropriate available drugs, a better understanding of the metabolic differences between different cancer cell types will also likely yield better disease outcome predictions. This is because some of the metabolic features of cancer cells are likely to be directly associated with disease aggressiveness and clinical outcomes[20-22].

Recently, comprehensive pathway curation projects (such as, for example, the Reactome and KEGG pathway projects) have successfully gathered high-quality information on human metabolic proteins and have accurately mapped these to metabolic pathways[23,24]. Cancer profiling projects such as that carried out by The Cancer Genome Atlas (TCGA) have yielded detailed genetic, transcriptomic, proteomic, and epigenetic data for thousands of human tumours each of which is annotated with clinical information for the patient from which it was taken[25]. Analysis of the TCGA data in the context of our present understanding of human metabolism should both illuminate the metabolic differences between different cancer types, and identify which of these differences has the most meaningful prognostic value. If this information is then coupled with the known drug responses of different cancer cell types, it should also be possible to identify the most suitable drugs to treat any particular cancer.

Valuable in this regard, are large-scale drug response screening projects such as the Genomics of Drug Sensitivity in Cancer (GDSC;[23]) and the Cancer Cell Line Encyclopedia[26] which provide transcriptome and epigenetic profiles for over one thousand human cancer cell lines together with their dose-response profiles to hundreds of anticancer drugs. The genetic, transcriptomic and epigenetic profiles of tumour samples from the TCGA and those of cancer cell lines from the GDSC can be directly compared to systematically test for metabolic similarities and differences that might have a bearing on drug responses. More specifically, the subset of the cancer cell lines that have genomic and transcriptomic features that are most similar to those of tumour cells from a patient could be used to interrogate how metabolic perturbations in the patient's tumour cells are likely to influence the effectiveness of particular anticancer drugs.

Here, we used data on gene mutations and copy number variations from the TCGA in conjunction with Reactome Pathways[25] data to identify the heterogeneous metabolic features of 32

human cancers. We then used these features together with drug response data from the GDSC to identify specific metabolic perturbations in tumour cells that are likely to impact their responses to different anticancer drugs.

## Results

We analysed a TCGA dataset comprising lists of gene alterations (mutations and copy number variations) together with clinical information collected from 10,528 patients afflicted by 32 different human cancers (Fig. 1a). Also, we analysed lists of gene alterations found within the genomes of 812 human cancer cell lines together with the drug-response profiles of these cell lines to 251 anticancer drugs to reveal associations between gene alterations and drug responses.

**Alterations to genes involved in metabolism distinguish human cancers.** We obtained curated human metabolic pathway data and the names of genes involved in these pathways from the Reactome Pathways database using the annotation search term "metabolism"[25]. In this database, the term "metabolism" encompasses 68 different metabolic pathways involving 2325 genes. Within the TCGA dataset, we found that out of these 2325 genes, 2095 contained an alteration in at least one of the 10,225 analysed patients.

Among the 2095 metabolic genes displaying some alteration (a copy number variation or a mutation) in at least one patient, the most frequently altered were *PIK3CA* in 1384 individual tumour samples, *APOB* in 976 and *LRP2* in 961 (Fig. 1b). Most of the genes displaying some alterations in tumours of different cancer types have well-defined roles in carcinogenesis. For instance, mutations of *PIK3CA* reprogramme metabolism and are associated with poorer survival outcomes in several cancers, including those of the colon, rectum, breast and lungs[26-29]. *APOB* is a lipid metabolism regulator that is linked to carcinogenesis and tumour progression in the liver, lungs and other tissues[30-32]. *LRP2* encodes a low-density lipoprotein receptor-related protein-2 which mediates endocytic uptake of various lipids, and is linked to the enhanced metabolism of lipids and vitamin D, and promotes the transformation, proliferation and survival of various types of cancer cells[33-35].

Next, we calculated the frequency of alterations among the 16 first-tier metabolic pathways across all 32 of the cancer types. Here we found that genes involved in lipid metabolism were the most commonly altered, followed by those involved in carbohydrate metabolism and then those involved in amino acid metabolism (Fig. 1c). These findings echo the well-established tenet of molecular oncogenesis, that meeting the cellular energy and biosynthetic demands of malignancy require alterations to the lipid, carbohydrate and amino acid metabolic pathways[3,19].

We clustered the 32 human cancers based on the frequencies of gene alterations of metabolic pathways. Our clustering revealed two major groups of cancers (Supplementary Fig. 1): those cancers with a higher frequency of metabolic gene alterations (which we named as HM; $n = 6,191$) and those with a lower frequency of metabolic gene alterations (named as LM; $n = 3,329$). Interestingly, we observed that the median alteration frequencies for genes involved in each of the 16 first-tier metabolic varied across the 32 cancer types, e.g., occurring in 90% of patients with skin cutaneous melanoma, but only 14% of patients with thyroid carcinomas (Fig. 1c).

We examined whether the HM and LM cancer supertypes were associated with different clinical outcomes. Remarkably, we observed that the median disease-free survival (DFS) periods was significantly lower ($p = 1.3 \times 10^{-7}$; log-rank test[36]) for the HM cancer patients (median = 58.3 months) than it was for the LM cancer patients (median = 116.2 months; Fig. 2a). Similarly,

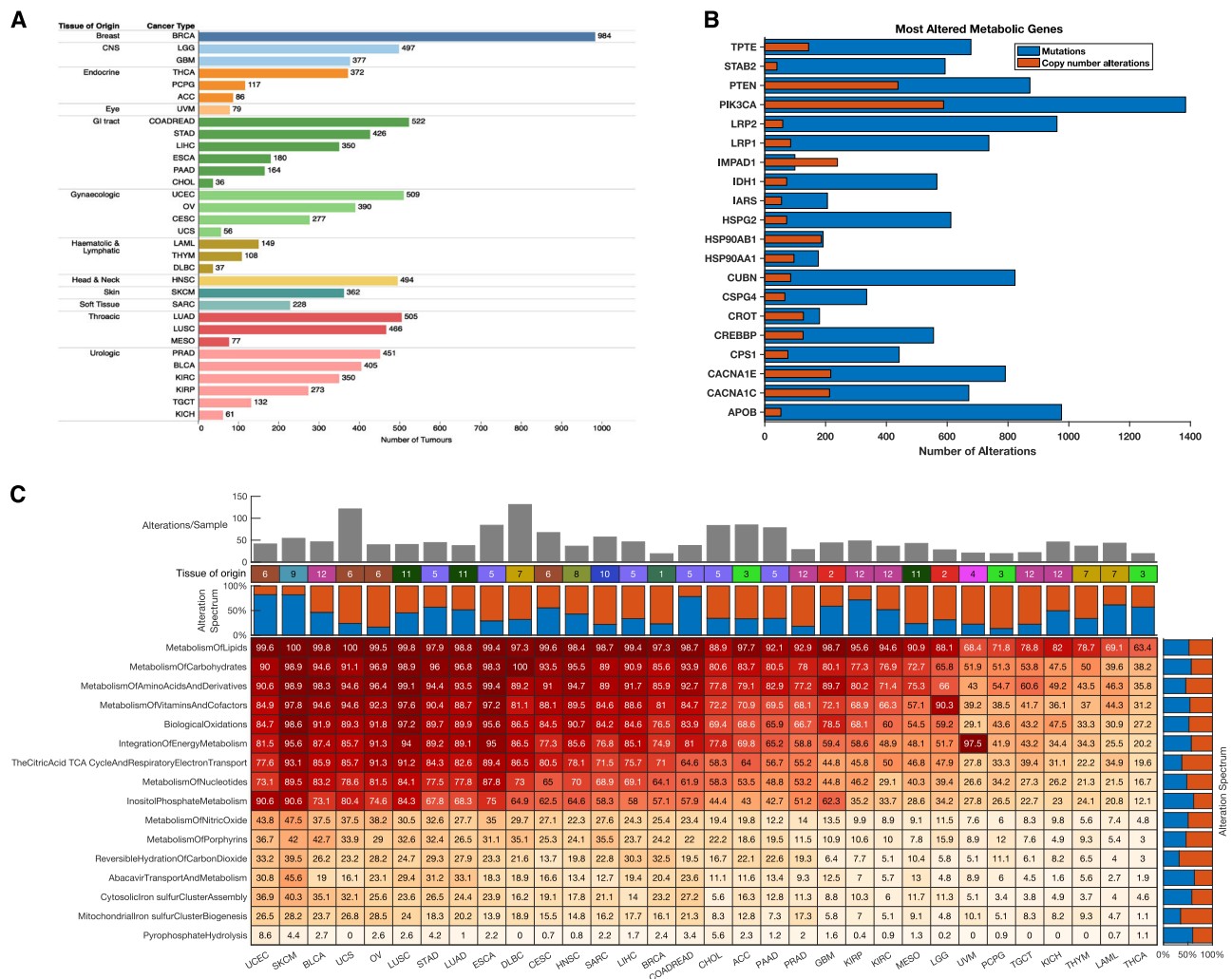

**Fig. 1 a** Distribution of 10,528 TCGA tumours across 32 human cancer types broken down by tissue of origin. TCGA disease codes and abbreviations: UCEC, uterine corpus endometrial carcinoma; SKCM, skin cutaneous melanoma; BLCA, bladder urothelial carcinoma; UCS, uterine carcinosarcoma; OV, ovarian serous cystadenocarcinoma; LUSC, lung squamous cell carcinoma; STAD, stomach adenocarcinoma; LUAD, lung adenocarcinoma; ESCA, oesophageal adenocarcinoma; DLBC, diffuse large b-cell lymphoma; CESC, cervical squamous cell carcinoma; HNSC, head and neck squamous cell carcinoma; SARC, sarcoma; LIHC, liver hepatocellular carcinoma; BRCA, breast invasive carcinoma; COADREAD, colorectal adenocarcinoma; CHOL, cholangiocarcinoma; ACC, adrenocortical carcinoma; PAAD, pancreatic adenocarcinoma; PRAD, prostate adenocarcinoma; GBM, glioblastoma multiforme; KIRP, kidney renal papillary cell carcinoma; KIRC, kidney renal clear cell carcinoma; MESO, mesothelioma; LGG, brain lower grade glioma; UVM, uveal melanoma; PCPG, pheochromocytoma and paraganglioma; TGCT, testicular germ cell tumours; KICH, kidney chromophobe; THYM, thymoma; LAML, acute myeloid leukaemia; THCA, thyroid carcinoma. **b** Genes involved in metabolism found to be most altered across all human cancers. **c** Clustered heatmap of cancer types using the percentage of tumours with first-tier metabolic pathway genes displaying alterations. Pathways are ordered by decreasing frequencies of alterations. Increasing colour intensities denote higher percentages. The heat map was produced using unsupervised hierarchical clustering with the Euclidean distance metric and complete linkage (see Supplementary Fig. 1). The coloured bars on the heatmap show the tissue of origin for each cancer: 1 = Breast; 2 = CNS, 3 = Endocrine; 4 = Eye; 5 = GI tract; 6 = Gynaecologic; 7 = Haematologic & Lymphatic; 8 = Head & Neck; 9 = Skin; 10 = Soft Tissue; 11 = Thoracic; 12 = Urologic. The bar graph represents the overall frequency of genomic alterations in each human cancer

the duration of overall survival (OS) periods for the HM cancer patients (OS = 68.9 months) were significantly shorter ($p = 6.8 \times 10^{-10}$) relative to those of the LM cancer patients (OS = 116.2 months; log-rank test; Fig. 2b). We validated these findings with an independent dataset of patients afflicted with these cancers from the International Cancer Genome Consortium (ICGC) databases[37]. As with the patients recorded in TCGA, the median OS period for patients recorded in the ICGC databases who had cancers belonging to the HM supertype (OS = 1,759 days) was significantly lower ($p = 6.3 \times 10^{-17}$) than that of patients with cancers belonging to the LM supertype (OS = 3,681 days; Fig. 2c). Our results, therefore, demonstrate an association between the extent to which metabolic genes in cancer

cells are altered (and therefore probably the degree of metabolic dysregulation within these cells), and the aggressiveness of cancers.

## Alterations of genes involved in carbohydrate, amino acid and lipid metabolic pathways across all cancers

We evaluated the extent of alterations to genes involved in second-tier lipid, carbohydrate and amino acid metabolic pathways as these pathways had the highest gene alteration frequencies across all 32 of the cancer types. We found that alterations to genes involved in second-tier pathways were more frequent in the HM cancers than in the LM cancers (Fig. 3). Among the genes involved in second-tier

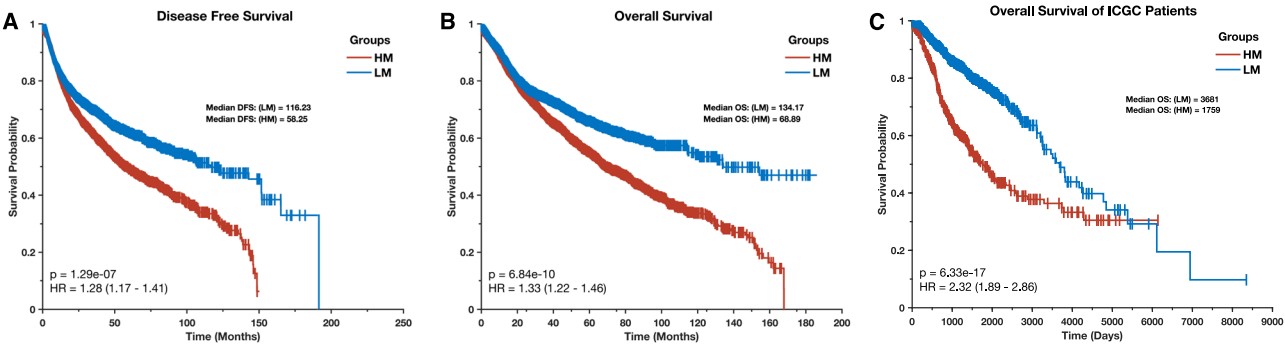

**Fig. 2** Kaplan–Meier curve of the disease-free survival periods (**a**) and overall survival periods (**b**) of TCGA patients afflicted by the HM (high metabolic gene alteration frequencies) and LM (low metabolic gene alteration frequencies) cancer supertypes. **c** Kaplan–Meier curve of the overall survival periods of ICGC patients afflicted by the HM and LM cancer supertypes

**Fig. 3** Frequency of tumours of different cancer types with altered genes that are involved in second-tier metabolic pathways of carbohydrate, lipid and amino acid metabolism. The cancers are arranged according to how they clustered based on similarities between their first-tier metabolic pathway gene alterations (as in Fig. 1c). Increasing colour intensities denote higher percentages of tumours with gene alterations

carbohydrate metabolism pathways, those involved in the glycosaminoglycan metabolism (in 67% of all patients' tumours) and glucose metabolism (in 58% of tumours) pathways were the most commonly altered across all cancers. In recent years, cellular glycosaminoglycan profiles have been shown to be markedly altered during tumour pathogenesis and progression. Glycosaminoglycans influence cell signalling, angiogenesis, tumour invasiveness and metastasis, and have therefore emerged as essential pharmacological targets for the treatment of cancer[38–40].

Among the genes involved in second-tier amino acid metabolism pathways, those involved in selenoamino acid metabolism (in 56% of all patients' tumours) and polyamine metabolism (in 56% of tumours) were the most altered across all the cancer types. Increased polyamine metabolism is associated with neoplasia: an important risk factor for the development of cancer in humans[41–45]. Drugs that target polyamine metabolism, several of which are in clinical trials, have been considered for the treatment of many cancers, including those of the colon, prostate and skin[41,42,46]. Unlike with polyamines, the roles of selenoamino acids in cancer

remain poorly explored; although an enrichment of selenoamino acids has been noted in breast cancer cells[47]. We anticipate that studying alterations of selenoamino acid metabolism could yield targets for the development of new therapeutics and predictive biomarkers that would aid the treatment of various cancers.

Abnormal lipid metabolism has emerged as a metabolic hallmark of oncogenesis and tumour progression[48]. Here, we found that across all cancers, the most frequently altered of the lipid metabolism genes were those involved in the phospholipid metabolism (in 79% of all patients' tumours) and fatty acid metabolism (in 68%). Changes in the transcripts of genes that encode membrane phospholipids and the actual levels of phospholipids have been shown in various cancers, including those of the breast and lung[49–51]. Since the changes in phospholipid metabolism can affect the proliferation of cancer cells and their responses to drugs, it is plausible that at least some of the observed alterations in genes involved in phospholipid metabolism may have biological and clinical relevance[51,52].

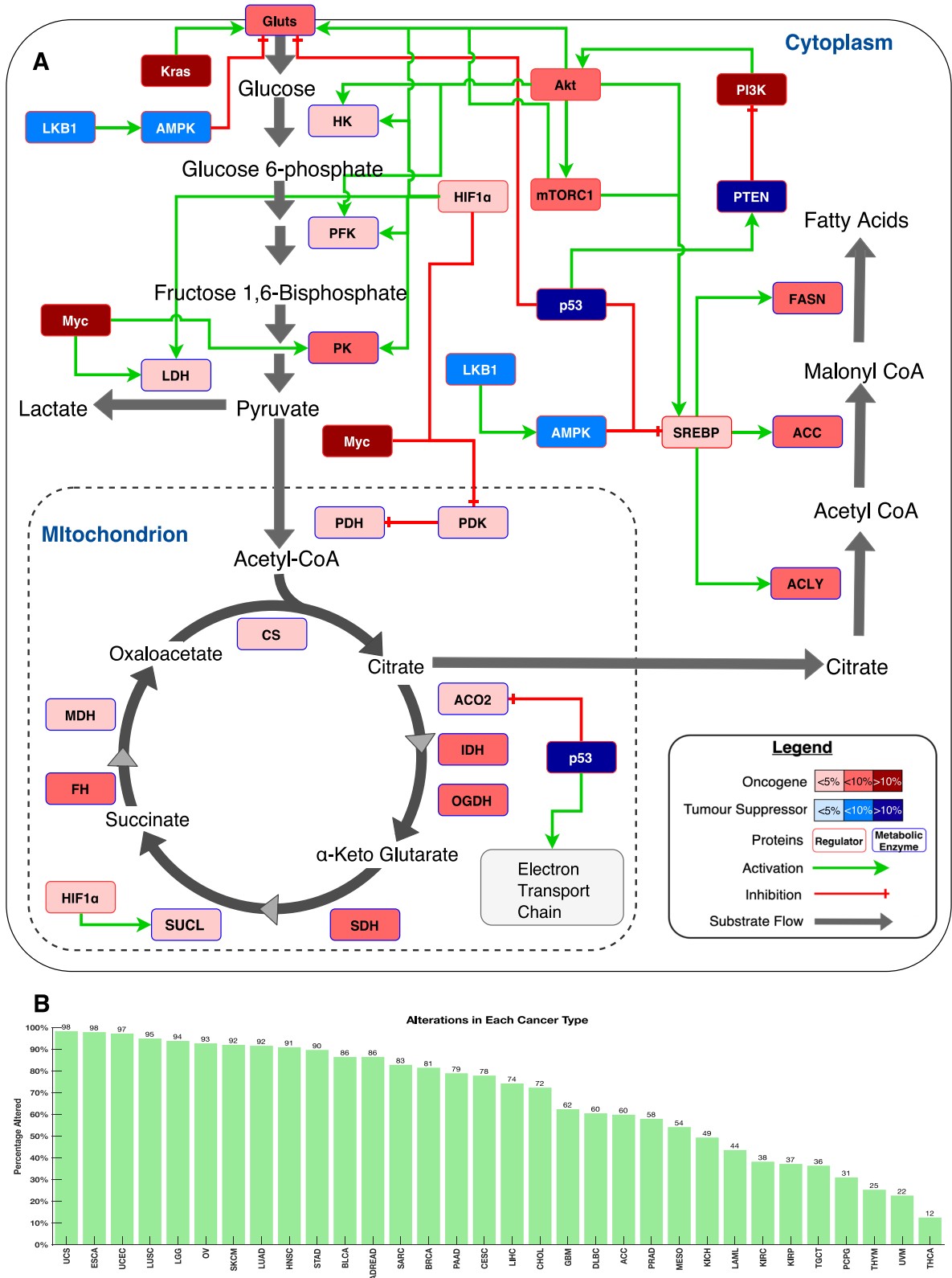

Figure B: Alterations in Each Cancer Type

Some of the most studied metabolic pathways in cancer are the glycolytic and fatty acid oxidation and biosynthesis pathways[1,15–18,53]. Here, we also explored the degree to which genes that are involved in these pathways were altered in each of the 32 cancers. In all cancers, we found alterations to some of the genes involved in the glycolytic and fatty acid oxidation and biosynthesis pathways (Supplementary Figs. 2, 3, 4 and 5). We

found that these gene alterations were most frequent in uterine corpus endometrial carcinomas and skin cutaneous melanomas.

Finally, we used the literature to identify a subset of genes that encode proteins which are either key metabolic enzymes of the central metabolic pathways or are regulators of these enzymes. We discovered that 78% of all tumours harbour alteration in these genes (Fig. 4a). Among the most frequently altered metabolic

**Fig. 4 a** Major catabolic and anabolic pathways of glucose and lipid metabolism in human cells. Nodes represent either enzymes (blue outline colour) or metabolic regulators (red outline colour). Node colours represent tumour suppressors (blue) and oncogenes (red) and their increasing colour intensities denote higher percentages of tumours with alterations in the genes encoding these enzymes or regulatory proteins. Edges indicate known types of interaction: red for inhibition and green arrows for activation. Abbreviations: GLUTs, all glucose transporters; HK, hexokinase; PFK, phosphofructokinase; PK, pyruvate kinase; LDH, lactate dehydrogenase; PDH, pyruvate dehydrogenase complex; PDK; pyruvate dehydrogenase kinase; CS, citrate synthase; ACO2, cis-aconitase; IDH, isocitrate dehydrogenase; OGDH, α-ketoglutarate; SDH, succinate dehydrogenase; SUCL, succinyl-CoA lyase; FH, fumarate hydratase; MDH, malate dehydrogenase; ACLY, ATP-dependent citrate lyase; ACC, acetyl-CoA carboxylase; FASN, fatty acid synthase; PTEN, phosphatase and tensin homologue; AMPK, 5'-AMP-activated protein kinase; mTORC1, mechanistic target of rapamycin complex-1; PI3K, phosphoinositide-3 kinase; SREBP, Sterol regulatory element-binding protein; Akt, RAC-alpha serine/threonine-protein kinase; Kras, Kirsten rat sarcoma viral oncogene homologue; Myc, MYC proto-oncogene; HIF1α, hypoxia-inducible factor 1-alpha; LKB1, Liver Kinase B1; p53, p53 tumour suppressor. **b** overall fraction of samples with the central metabolic pathways gene alterations across 32 human cancers

regulators were *PTEN* (in 14% of all tumours), *KRAS* (in 11%) and *MYC* (in 11%). These gene alterations were most frequent in uterine carcinosarcoma (98.2% of patients' tumours) and least frequent in thyroid carcinomas (in 12.4% of tumours; Fig. 4b).

Collectively these results reiterate that alterations within genes involved in particular aspects of lipid, carbohydrate and amino acid metabolism are found in many different cancers.

**Alterations of genes involved in metabolism are associated with alterations of mRNA transcript levels.** We next determined whether alterations in genes that are involved in metabolism are associated with alterations to the encoded mRNA transcript levels of these genes. We first examined whether the HM and LM cancer supertypes displayed distinct mRNA signatures for genes involved in metabolic pathways. Among the 2325 genes involved in metabolism, we found that only 1977 genes had transcript measurements in the TCGA. Therefore, focusing only on these 1977 mRNA transcripts in all patients afflicted with the 32 different cancers, we applied t-distributed stochastic neighbourhood embedding (t-SNE) to reduce the dimensions of these data and visualised the relationships between cancers using scatter plots. We found that whereas the HM cancers displayed similar patterns of mRNA expression (i.e. they clustered closer to one another in the scatter plots; Fig. 5a), the LM cancers tended to display more diverse mRNA expression patterns (i.e. they did not cluster as much in the scatter plots; Fig. 5b). Specifically, whereas a three-dimensional t-SNE plot indicated that the HM cancers tended to group in the centre of the gene expression space, the LM cancers were scattered around the periphery of this space (Fig. 5c).

Since the HM cancers tended to cluster together, we hypothesised that their metabolic gene expression profiles were highly correlated. To test this hypothesis, we measured the Pearson's linear correlation coefficients between transcript abundances across each pair of the 32 human cancers (see methods section). Indeed, we establish that whereas the mRNA transcript levels of the 1977 metabolic genes of each pair of HM cancers tended to be strongly positively correlated (mean Pearson's correlation = 0.9; range: 0.79–0.98), there tended to be weaker positive correlations between the mRNA transcript levels seen between the LM cancers (mean Pearson's correlation = 0.68; range: 0.40–0.92; Fig. 5d).

Overall, these results indicate that while gene expression profiles are relatively conserved among the HM cancers, they are more diverse in the LM cancers.

Since the relative uniformity of the HM group was intriguing, we decided to further evaluate tumours in this supertype using data on all 20,502 of the mRNA transcripts that are available in the TCGA database (i.e., not only the transcript of metabolic genes). Here, we applied t-SNE to visualise the grouping of HM tumours (Fig. 6a) and also applied Density-based spatial clustering of applications with noise (DBSCAN;[54,55]) approach

to classify the tumour into various subgroups (Fig. 6b). We found that patients afflicted with the different subgroups of tumours identified using DBSCAN exhibited different durations of DFS (Fig. 6c) and OS (Fig. 6d).

**Gene expression and enrichment characteristics of the HM and LM cancer supertypes.** We established that the transcripts which were differentially expressed between the supertypes were predominantly involved in a variety of different signalling pathways (see Supplementary File 3). Compared with HM tumours, LM tumours displayed elevated transcription levels of genes involved in, among other things, molecular functions associated with potassium channel activity, transmitter-gated ion channel activity, and sodium channel activity (Supplementary Fig. 6, also see Supplementary file 3). Alternatively, HM tumours displayed elevated transcription levels of genes involved in, among other things, the functions associated with endopeptidase inhibitor activity, alcohol dehydrogenase activity and oxidoreductase activity (Supplementary Fig. 6, also see supplementary file 3).

**The drug responses of cancer cell lines are associated with metabolic gene alterations.** From the GDSC database, we collected gene alteration data for 812 cancer cell lines of 30 different human cancer types, which also have dose-response profiles to 251 anticancer drugs (Fig. 7)[23]. We assessed the patterns of metabolic gene alterations within these cancer cell lines and discovered that these were similar to those of the primary tumours (Fig. 8a and Supplementary Fig. 6).

Given that previous studies have underlined differences in molecular characteristics between cancer cell lines and their primary tumour tissues[56,57], we directly compared metabolic gene alterations between cell lines and tumours of the same type. This revealed that, with only two exceptions, there were no significant differences in the frequencies of metabolic gene alterations between the cell lines and primary tumours of a given cancer type. The two exceptional cases were acute myeloid leukaemia ($\chi^2 = 22.7$, $p = 1.9 \times 10^{-6}$) and thyroid carcinoma ($\chi^2 = 16.7$, $p = 5 \times 10^{-4}$) for which the cell lines have significantly higher frequencies of metabolic gene alterations than did primary tumours (Fig. 8a; Supplementary Fig. 7, Supplementary file 1).

Next, we classified the cancer cell lines into either the HM or LM supertypes using the TCGA cancer type labels of each cell line that are provided within the GDSC database. We then compared drug IC50 values between HM and LM cell lines for 24 classes of drugs that target 24 signalling pathways and/or biological processes (Supplementary Fig. 8). Remarkably, we uncovered differences between the HM and LM cancer cell lines in their observed dose-responses to various classes of anticancer drugs. Compared to the HM cell lines, the LM cell lines were more sensitive to seven out of the 24 classes of anticancer drugs (Fig. 8b; Supplementary file 2). Surprisingly, the HM cell lines were only more sensitive than the LM cell lines to drugs that

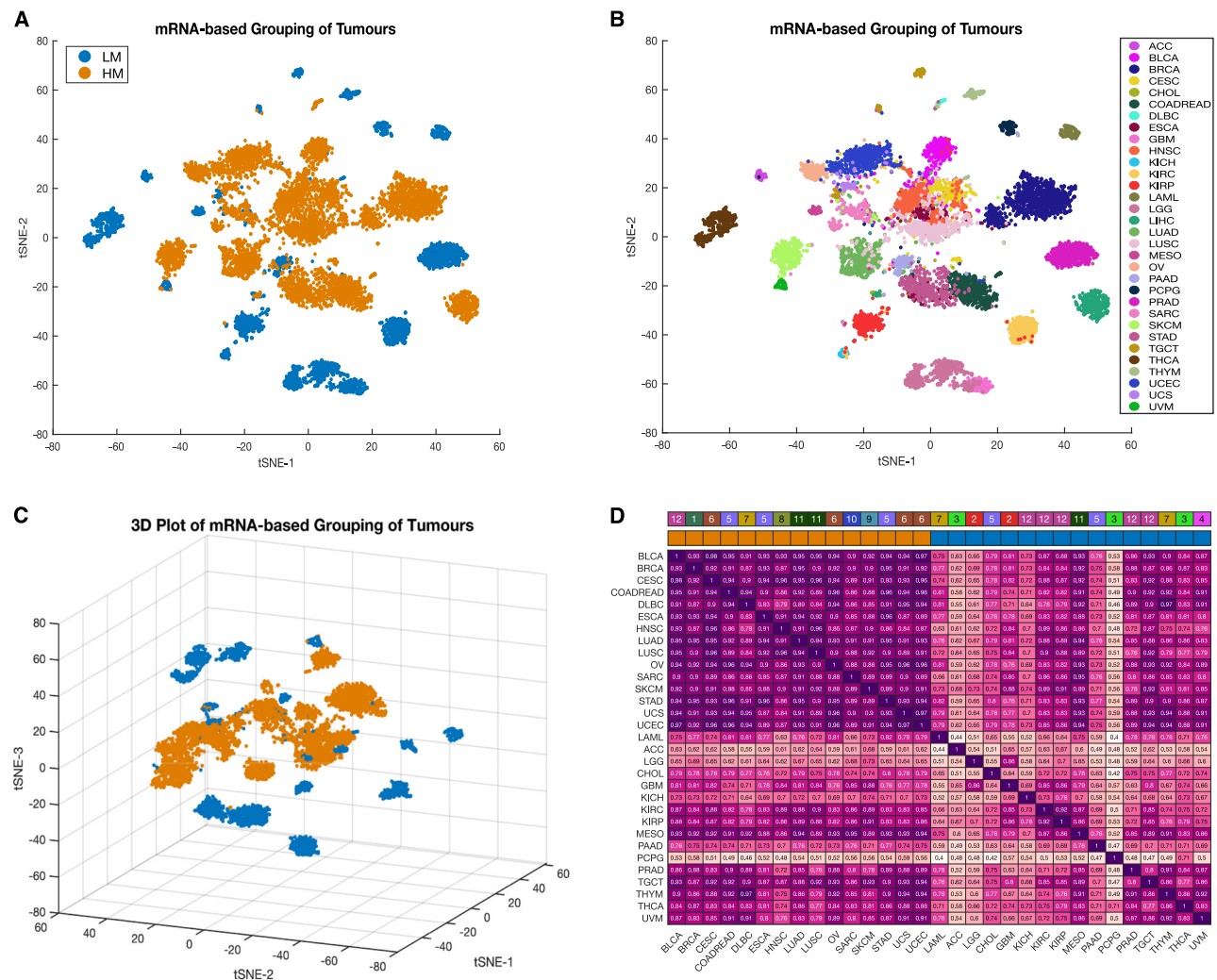

**Fig. 5 a** Clustering of HM (orange points) and LM (blue points) tumours based on mRNA transcript levels. **b** Clustering of 32 different cancer types based on mRNA transcript levels. Points are coloured according to the type of cancer they represent. For both plots (**a** and **b**), t-SNE was used to visualise the tumour classes using the exact algorithm and standardised Euclidean distance metric. **c** Three-dimensional plot of the HM/LM tumour supertype grouping based on mRNA transcript levels. **d** The integrated plot of mRNA expression correlations ordered by whether cancers belong to the HM or LM supertypes. From top to bottom, panels indicate: the tissue of origin; whether tumours belong to the HM or LM supertype; heatmap of inter-tumour linear Pearson's correlation scores with increasing colour intensities denoting higher degrees of correlation

target the EGFR signalling pathway (Fig. 8b; Supplementary file 2).

We next compared the IC50 values of all 251 individual drugs with which the LM and HM cell lines were treated, regardless of the drugs' modes of action. Here, we found that, after correcting for multiple comparisons, the IC50 values of 41 anticancer drugs differed significantly between the LM and HM cell lines (Supplementary file 2). Interestingly, the HM cell lines were more sensitive to only five of these 41 drugs. These included afatinib ($p = 2.1 \times 10^{-9}$), CP724714 ($p = 5.5 \times 10^{-4}$), gefitinib ($p = 6.3 \times 10^{-4}$), TAK-715 ($p = 0.02$), and vinorelbine ($p = 0.049$). Among these, afatinib, CP724714 and gefitinib target the EGFR signalling pathway, whereas TAK-715 targets JNK and p38 signalling, and vinorelbine inhibits mitosis by destabilising microtubules (Supplementary file 2). Conversely, we observed that the LM cell lines were significantly more sensitive than the HM cell lines to 36 of the anticancer drugs including CHIR-99021 ($p = 4.6 \times 10^{-7}$), QL-XI-92 ($p = 4.6 \times 10^{-7}$) and SN-38 ($p = 9.2 \times 10^{-5}$; see Supplementary file 2).

Overall this indicates that frequencies of metabolic gene alterations (our exclusive criterion for placing cell lines into the

LM and HM supertypes) is a highly relevant variable when attempting to predict the drug responsiveness of cell lines and, therefore, that it may also be a clinically relevant variable when predicting the drug responsiveness of primary tumours.

**The subtypes within each cancer exhibit diverse responses to anticancer drugs**. For each of the 32 TCGA cancer types, we applied unsupervised hierarchical clustering to counts of alterations within genes involved in the 16 first-tier metabolic pathways to identify disease subtypes within each cancer type (see examples in Supplementary Fig. 9). Here, we found that (aside from genes involved in lipid, carbohydrate, and amino acid metabolism) subgroups of patients are likely to harbour additional alterations in other metabolic pathways. For example, in the case of glioblastoma multiforme (Supplementary Fig. 9A) and lung adenocarcinoma (Supplementary Fig. 9B), we found that while almost all the tumours represented in TCGA have alterations to genes involved in lipid, carbohydrate and amino acid metabolism pathways, small groups of tumours usually show even higher numbers of alterations to genes

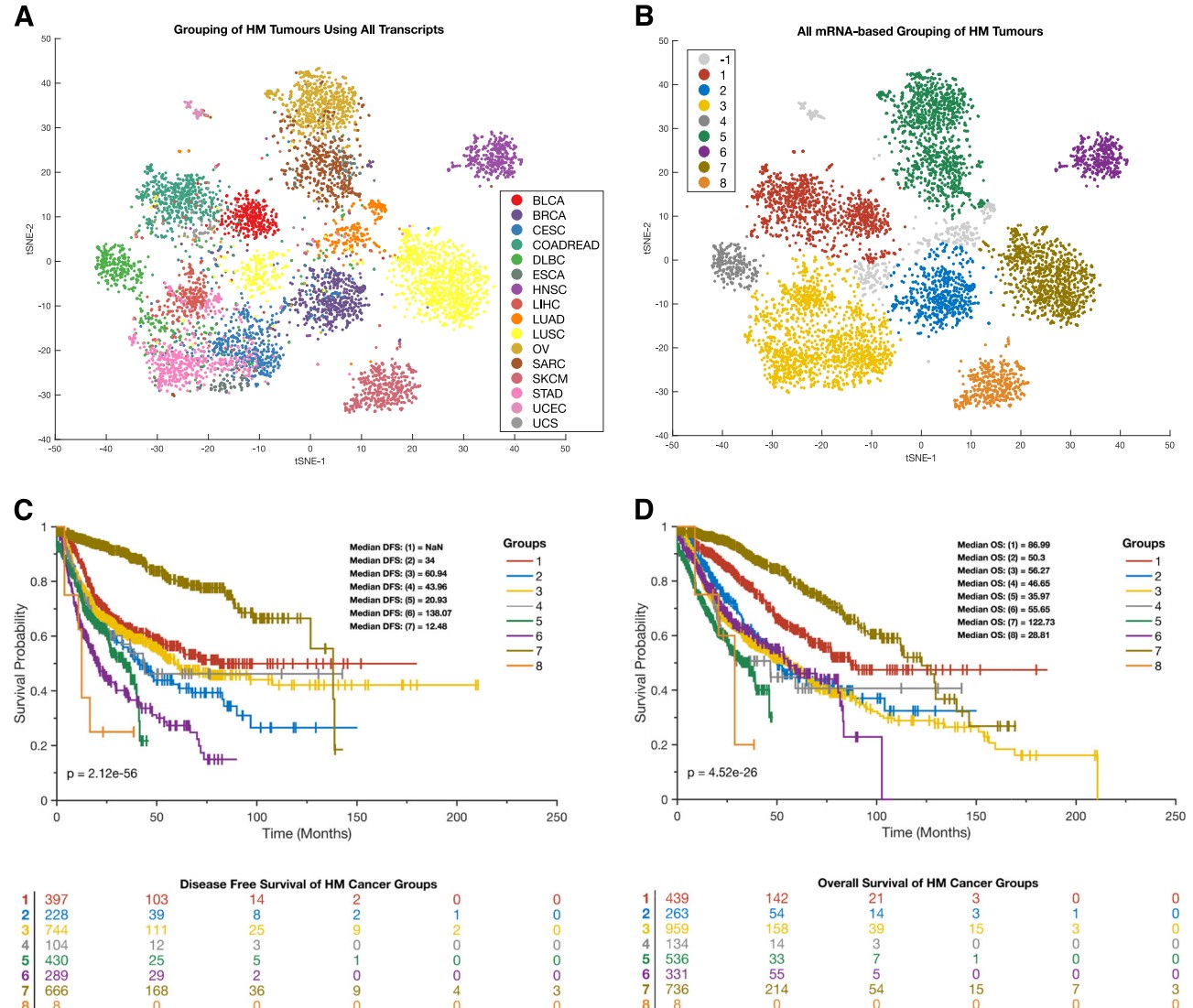

**Fig. 6 a** Clustering of HM tumours based on all 20,502 mRNA transcript levels that were measured by the TCGA project. The colour legend represents different cancer types. **b** Clustering of HM tumours based on all 20,502 mRNA transcript levels that were measured by the TCGA. Points are coloured according to the clustering of the tumour using DBSCAN. -1 indicates the outlier points. For both plots (**a** and **b**), t-SNE was used to visualise the tumour classes using the exact algorithm and standardised Euclidean distance metric. **c** Kaplan–Meier curve of the disease-free survival periods and the life table of patients afflicted with each DBSCAN disease subtype. **d** Kaplan–Meier curve of the overall survival periods and life table of patients afflicted with each DBSCAN disease subtype. For both survival curve plots (**c** and **d**), the colours represent the tumour groupings yielded by DBSCAN in panel B

involved in, amongst others, the abacavir and nitric oxide metabolism pathways.

Since the frequencies of alterations to genes involved in metabolic pathways are likely to influence the responses of patients to anticancer drugs, we identified GDSC cancer cell lines displaying similar gene alterations to those found in individual primary tumours to test whether this might be the case (see methods section). Here, we applied an approach were, for all cell lines of a particular human cancer, we compared their IC50 values for each of the 251 anticancer drugs between the cell lines with or without alterations to genes involved in each of the 16 first-tier metabolic pathways. Interestingly, we found that for cell lines of a particular cancer type, there are gene-alteration-dependent differences in their dose-responses to various anticancer drugs (Supplementary file 2). For example, 51 anticancer drugs demonstrated higher efficacies on oesophageal adenocarcinoma cell lines that have alterations in genes involved in abacavir metabolism pathways than on oesophageal adenocarcinoma cell

lines without alterations to these genes (Fig. 9a). Also, cell lines of lung adenocarcinoma with alterations in genes involved in the biological oxidation pathways are more resistant to 52 anticancer drugs than are those without alterations to these genes (Fig. 9a).

Altogether, we found 2186 instances where alterations to genes involved in a specific metabolic pathway are associated with the efficacy of anticancer drugs in the cancer cell lines (Supplementary file 2). Among the metabolic pathways, we found that those of cytoplasmic iron-sulphur clusters (447 instances), nucleotide metabolism (292 instances), and amino acid and derivatives metabolism (293 instances; Fig. 9a) were associated with varied efficacies of the highest numbers of anticancer drugs for all the cancer cell lines across all tumour types.

Given that we had found that tumours displaying different numbers of alterations to metabolic genes exhibit different clinical and survival outcomes, we decided to examine this in more detail for particular cancer types. Using data of primary cancers from the TCGA, for patients' tumours with or without

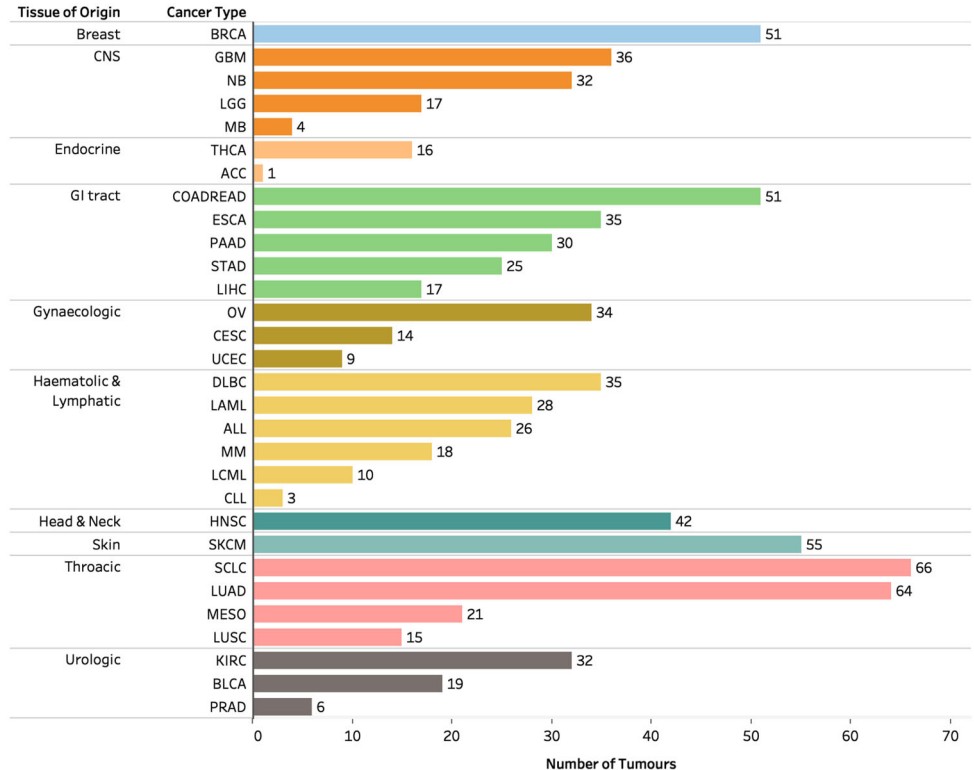

**Fig. 7** Distribution of 1001 cancer cell lines derived from 32 human cancer types broken down by tissue of origin

alterations in genes involved in abacavir metabolism, we found that the durations of the disease-free progression periods were significantly lower for oesophageal adenocarcinoma patients with alterations to these genes (log rank $p = 0.004$; Fig. 9b). Conversely, disease-free progression periods were higher for uterine corpus endometrial carcinoma patients with alterations to genes involved in abacavir metabolism (log rank $p = 0.041$; Fig. 9c). This then indicates that, even within each cancer type, the numbers of alterations found in metabolic genes involved in particular pathways can, in addition to influencing anticancer drug responses, detectably impact patient survival.

**Discussion**

We examined the relationships between the numbers of alterations within the metabolic genes of primary tumours and cell lines of 32 different human cancer types and both clinical outcomes and likely drug responses. Others have used mRNA transcript data to show that alterations in metabolic pathways likely differ substantially between human cancer types[7,8]. To the best of our knowledge, ours is the first study to characterise metabolic gene alterations across such a large number of primary tumours (10,528) for so many distinct cancer types (32).

While we found at least one altered metabolic gene in every one of the 10,528 analysed tumours, the numbers of altered metabolic genes varied between the 32 cancer types that these tumours belonged to. We demonstrated that a clinically relevant clustering of patient tumours, irrespective of the type of cancer they represented, could be achieved by simply dividing the tumours into two supertypes based entirely on the numbers of alterations they displayed in metabolic genes: an LM supertype for low numbers of metabolic gene alterations and an HM supertype for high numbers of metabolic gene alterations (Supplementary Fig. 1). Just as others have shown that alterations of genes involved in signalling pathways can have clinical implications[58,59], we show here that individuals with HM tumours tend to have worse clinical outcomes

than those afflicted with LM tumours. As such, our results suggest that simple counts of metabolic gene alterations in a tumour can provide a quantitative approximation of the extent of metabolic dysregulation within the tumour and, hence, an indirect approximation of the aggressiveness of the tumour.

Our analyses indicate that alterations of genes involved in the central metabolic pathways and the regulators of these pathways are pervasive across all human cancers (Fig. 4). Among the most commonly altered of the regulatory genes that are involved in cellular metabolism were *PIK3CA* (in 32% of tumours), *MYC* (in 14%) and *HIF1A* (in 11%). In various cancers, *MYC* and *HIF1A* alterations dysregulate multiple metabolic enzymes including, hexokinase, isocitrate dehydrogenase, pyruvate dehydrogenase kinase and lactate dehydrogenase[60,61]. Further, *PIK3CA*, *MYC*, *HIF1A* and other genes with frequent alterations in primary tumours are known to dysregulate cellular metabolism by increasing the rate of glycolysis while reducing the rate of aerobic respiration; a phenomenon referred to as the Warburg effect[2,60,62]. Tumours that exhibit a Warburg phenotype are known to be more aggressive and respond more poorly to most anticancer drugs[63]. Accordingly, compared to the LM cancers, we found higher alteration rates of the Warburg phenotype-associated genes in the HM cancers, which could explain why patients afflicted with HM cancers tend to have worse survival outcomes.

Changes in various signalling pathways are associated with variations in the response of cancer cells to drug perturbations, and these changes can, therefore, impact disease treatment outcomes[64,65]. Prior to the provision of anticancer drugs, it is desirable to know the drugs to which a particular tumour is most likely to be responsive. Since it is practically impossible to test hundreds of individual drugs on a specific tumour, cell lines that have phenotypic features resembling that of the tumour may be useful in predicting the drug responses of that tumour[23,66–68]. Accordingly, using drug response data for cancer cell lines, we

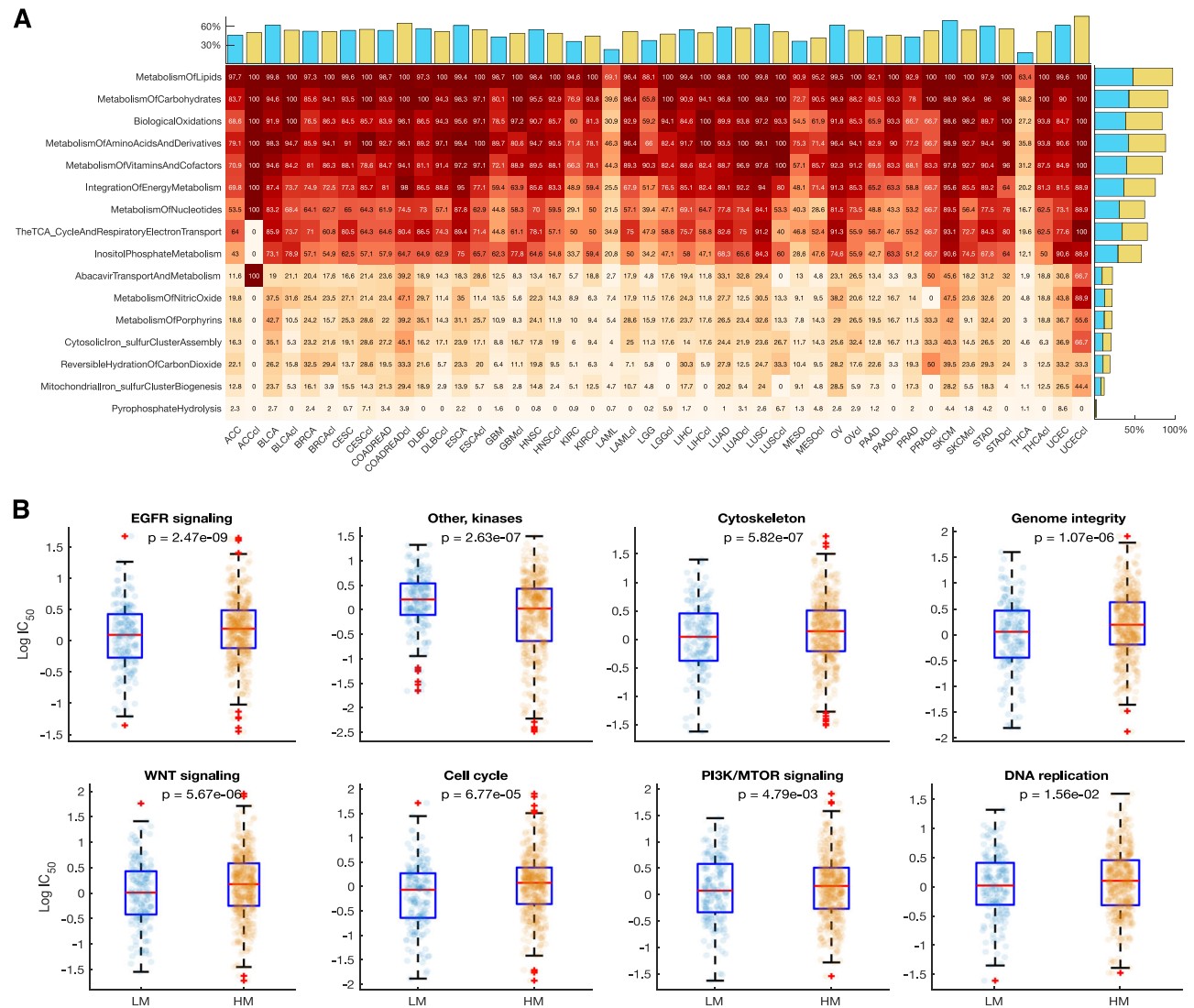

**Fig. 8 a** Heatmap of the fraction of altered GDSC cancer cell line genes that are involved in each first-tier metabolic pathway in relation to corresponding patient tumour data from TCGA. Pathways are ordered according to numbers of observed alterations within genes that are involved in the pathways. Increasing colour intensities denote higher percentages of tumours containing alterations in the genes involved in the represented pathways. Bar graphs above the heatmap indicate overall percentages of gene alterations within GDSC cell lines (blue bars) or TCGA tumours (tan bars) of a particular cancer type. Bar graphs on the right of the heatmap indicate the overall percentage of alterations within each first-tier metabolic pathway for the GDSC cell lines (blue bars) and TCGA tumours (tan bars). **b** Comparison of the dose-response profiles between the LM and HM supertypes of the GDSC cancer cell lines for selected drugs. Boxplots show the logarithm transformed mean IC50 values of the cancer cell lines that correspond to the HM and LM cancer supertypes. On each box, the central red mark indicates the median, and the bottom edge represents the 25th percentiles, whereas the top edge of the box represents 75th percentiles. The whiskers extend to the most extreme data points not considered outliers, and the outliers are plotted individually using the '+' symbol

inferred that HM and LM cancers are likely to respond differently to various anticancer drugs. Specifically, HM cancers tended to be less responsive to most anticancer drugs than LM cancers. This suggests that in addition to HM tumours potentially being more aggressive than LM tumours, patients afflicted with HM cancers may also exhibit worse clinical outcomes simply because HM cancers are more refractory to most anticancer drugs (Supplementary file 2). Also, since our results indicate that HM tumours are likely to only respond to higher doses of anticancer drugs, it would follow that patients with such tumours would tend to experience more adverse drug effects and treatment-associated complications, both of which could unfavourably impact their survival[69–72].

Drugs such as afatinib and gefitinib, which target the EGFR signalling pathway were, however, found to have higher efficacies in

HM cell lines than in LM cell lines. Currently, afatinib is the first-line treatment for patients with metastatic non-small cell lung cancer, and it has also been evaluated for the treatment of head and neck squamous cell carcinoma[73,74]. In our analyses, both non-small cell lung cancer and head and neck squamous cell carcinoma are HM cancers, and we predict, therefore, that there is a strong likelihood that many other HM cancers such as skin cutaneous melanoma, bladder urothelial carcinoma and lung adenocarcinoma may also respond to drugs that target the EGFR signalling pathway.

It is important to emphasise that our LM/HM classification is very simplistic. Taking a step back, we are reminded that among tumours that are derived from any particular tissue, there exist distinct tumour subtypes that differ from one another both in the gene alterations they display, and in the actual metabolic perturbations that these gene alterations cause[7,75–78]. In many

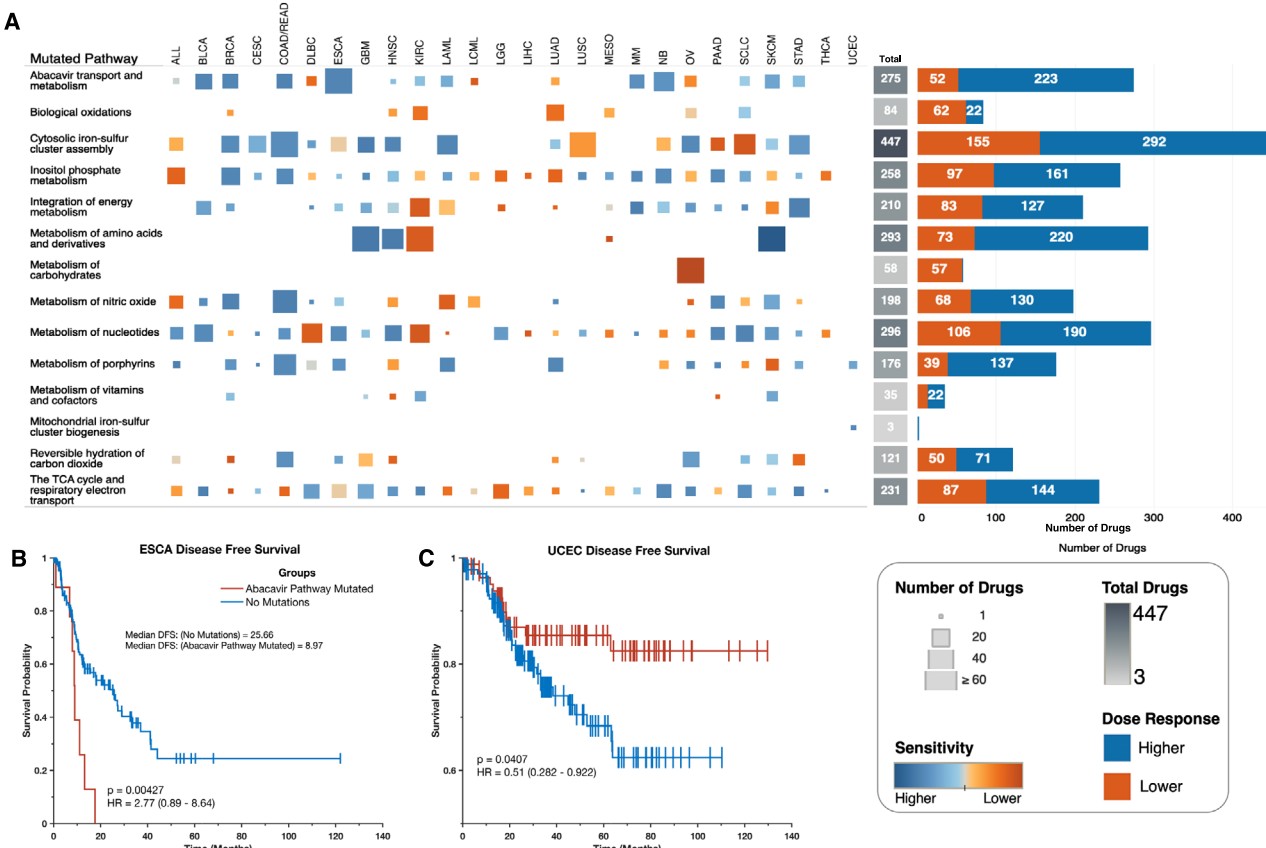

**Fig. 9 a** Dose-response profiles for drugs that have a degree of efficacy that is influenced by alterations in genes involved in specific metabolic pathways. From left to right: the columns represent GDSC cancer cell lines of various cancer types. The sizes of squares represent the number of drugs with efficacies that differ significantly between cell lines with and without gene alterations in the pathways indicated along the rows. The marks are coloured based on the overall influence of the metabolic gene alterations on drug efficacy: with increasing blue intensities denoting increasing sensitivity and increasing orange intensity denoting increasing resistance. The heatmap represents the overall numbers of drugs whose efficacy is influenced by the altered metabolic genes that are involved in the represented pathways. The bar graphs represent the total numbers of drugs whose dose-responses are increased (blue) or decreased (orange) by alterations of genes that are involved in the respective pathways. **b** Kaplan–Meier curve of the disease-free survival periods of patients afflicted with oesophageal adenocarcinoma with or without alterations to genes involved in the abacavir metabolism pathway. **c** Kaplan–Meier curve of the disease-free survival periods of patients afflicted with uterine corpus endometrial carcinoma, with or without alterations to genes involved in the abacavir metabolism pathway

respects, these distinct tumour subtypes are different ` diseases requiring different treatments[58,66,67].

We noted differences in the efficacy of various anticancer drugs between cell lines of the same primary cancer type. In some cases, these differences were associated with the presence or absence of alterations to genes involved in a particular metabolic pathway. This is in concordance with several recent studies that have established links between gene alterations and drug action[23,58,68,89]. This then supports the assertion that for any given cancer patient, the overall landscape of metabolic gene alterations could be used to identify generally applicable anticancer drug classes, following which alterations to specific metabolic genes could be used to eliminate the remaining drug choices that have the highest chances of failure.

Our results have revealed that within each of the 32 cancer types, there exist subtypes that have alterations in genes that are involved in metabolic pathways that are less commonly associated with cancers (Supplementary Fig. 9). Interestingly, we found that for different cancer types, alterations of genes involved in a particular metabolic pathway may not produce similar clinical outcomes. For example, we found that for patients with alterations to genes involved in abacavir metabolism, those afflicted with oesophageal adenocarcinoma present with worse outcomes whereas those afflicted with uterine corpora endometrial

carcinoma present with better outcomes (Fig. 9b, c). Such a scenario has been shown in other cancers. For instance, activation of the mitogen-activated kinase pathway is associated with worse clinical outcomes in ovarian and colorectal cancer[79,80], but with better clinical outcomes in hormone receptor-negative breast cancer and astrocytoma[81,82].

Altogether, we have shown both that metabolic gene alterations which potentially dysregulate metabolic pathways are a pervasive phenomenon across all 32 of the investigated human cancer types, and that numbers of metabolic gene alterations are linked to treatment outcomes. Further, our analysis of the drug response profiles of well-characterised cancer cell lines suggests that alterations of genes of various metabolic pathways may also be predictive of drug responses. While we cannot guarantee that simply scoring gene alterations of particular metabolic pathways in patient tumours will reveal the best available treatment choices for these patients, it is apparent that such scores could nevertheless be leveraged to increase the probability of making a good treatment choice.

## Methods

We analysed a TCGA project dataset of 10,528 patient-derived tumours representing 32 distinct human cancers (see Fig. 1a)[75], obtained from cBioPortal[83] version 2.20 (http://www.cbioportal.org). The elements of the data that we used to

identify gene alterations were gene copy number counts and somatic mutations (point mutations and small insertions/deletions). We also used mRNA expression data and comprehensive deidentified clinical data for all the TCGA study participants.

**Metabolic gene alterations in the TCGA cancers**. We accessed information of all human metabolic pathways from the Reactome pathways database version 68[25]. Reactome pathways are arranged into several tiers with the Reactome term "metabolism" (Reactome ID: R-HSA-1430728), encompassing 68 different metabolic pathways (see https://reactome.org/PathwayBrowser/#/R-HSA-1430728). The first-tier pathways include sixteen curated metabolic pathways which involve 2325 genes.

For each of the 32 human cancers, we calculated the overall percentage of samples with mutations and/or copy number alterations in genes that belong to each of the sixteen first-tier metabolic pathway as defined in the Reactome pathway database (see the spreadsheet, "Metabolic Pathways - First Tier", of Supplementary file 2). This provided us with alteration frequencies for each metabolic pathway in each human cancer (Fig. 1c). We applied unsupervised hierarchical clustering with the squared Euclidean distance metric to these data to identify altered metabolic gene supertypes of human cancers (Supplementary Fig. 1). Based on the clustering dendrogram that this yielded, we identified two cancer supertypes, which for simplicity, we named as either HM or LM, for those that respectively displayed higher or lower numbers of first-tier metabolic pathway associated gene alterations. The clustering of tumours into the two supertypes was highly coherent, with a cophenetic correlation coefficient of 0.89 and a Spearman's rank correlation between the dissimilarities and the cophenetic distances of 0.9[54].

We extracted information relating to the genes that encode proteins of the second-tier metabolic pathways for only three of the first-tier pathways: those of carbohydrate, lipid and amino acid metabolism. Again, we used the approach for determining the extent of gene alterations (as described above) to calculate the fraction of tumours with alterations to genes involved in second-tier metabolic pathways across each cancer type (Fig. 3). Also, using the same approach, we calculated the fraction of tumours with alterations in the genes that encode enzymes of the central metabolic pathway and their regulators (Fig. 4).

**Analysis of mRNA expression profiles of metabolic pathway genes across cancers**. We collected mRNA expression data of the genes that were profiled by the TCGA. Among this mRNA transcript data, we found information on only 1977 genes out of the 2325 genes that are involved in metabolism. We used the t-Distributed stochastic neighbourhood embedding algorithm to minimise the divergence between the 1977-mRNA transcripts across cancers to return a two- and three-dimensional embedding of the 32 human cancers[84]. The overall structure of these transcript embeddings was visualised using scatter plots, first based on the cancers' metabolic supertypes (HM and LM) and second based on the cancer types (Fig. 5a–c). To test for correlations between the mRNA transcripts of human cancers, we first calculated the mean transcript levels of the 1,977 metabolic pathway genes for each cancer and then used these mean values to calculate pairwise Pearson's linear correlation coefficients between each pair of the 32 human cancers (Fig. 5d)[85].

We retrieved all 20,502 of the mRNA transcripts that were measured by the TCGA project across all cancer studies and applied t-SNE to visualise the clustering of HM tumours across a dimensional space (Supplementary Fig. 6A). Further, we applied DBSCAN[54] to cluster tumours belonging to the HM cancer supertype into various subgroups (Supplementary Fig. 6A).

**Gene expression and enrichment characteristics of the HM and LM cancer supertypes**. The differentially expressed genes between the cancer supertypes were identified using the Student t-test with unequal variance and with the Benjamin-Hochberg correction applied to p-values[86,87]. Further, we queried Enrichr with two lists of 803 and 1118 genes found to be upregulated in HM tumours and LM tumours, respectively, to return enriched Gene Ontology (GO) molecular functions for each supertype (see Supplement File 3)[88]. A custom MATLAB script was used to create an enrichment network based on the enriched GO-molecular function designations. This enrichment network was visualised in yEd (Supplementary Fig. 6).

**Alterations of metabolic genes in cancer cell lines**. We obtained mutation and copy number alteration data for 1,002 cancer cell lines and 224,202 dose-response profiles of these cell lines to 267 anticancer drugs from the Genomics of Drug Sensitivity in Cancer (GDSC) database version 7.0 (www.cancerRxgene.org)[23]. For downstream analyses, we focused on only the 812 cancer cell lines for which a complete set of gene alterations and drug response data was available.

Next, we calculated the frequencies of alterations in genes involved in the sixteen first-tier metabolic pathways in the cancer cell lines using the approach previously described for the 32 human cancers (Fig. 8a, Supplementary Fig. 7). Finally, we used $\chi^2$ tests to identify possible differences between the TCGA cancers and the GDSC cell lines concerning the alteration counts of genes involved in the first-tier metabolic pathways (see results in supplementary file 1).

**Dose-response characteristics of the LM and HM cancer cell lines**. From the list of 812 GDSC cancer cell lines, we returned only 653 cancer cell lines for which the GDSC have assigned a TCGA classification to the cell lines' primary cancer. Altogether, these 653 cell lines corresponded to only 23 of the 32 different human cancers profiled by the TCGA (Fig. 7). The GDSC treated these 653 cancer cell lines with 251 distinct anticancer drugs that target 24 different signalling pathways and biological processes (Supplementary Fig. 8).

We used Student t-tests to compare the mean differences in the logarithm transformed IC50 values between the HM and LM cell lines for each class of anticancer drugs that we segregated based on the target signalling pathway and/or biological process (Fig. 8b, also see Supplementary file 2). Additionally, we compared the mean differences in the logarithm transformed IC50 values between HM and LM cell lines for each anticancer drug separately (Supplementary file 2).

**Identification of metabolic disease subtypes for each cancer type**. For each of the 32 human cancer types, we calculated the frequency of alterations to genes involved in the 16 first-tier metabolic pathways. We then applied unsupervised hierarchical clustering to these data to identify subtypes of disease for each cancer (see examples in Supplementary Fig. 9).

**Comparison of dose-response profiles within each cancer type for tumours with or without specific pathway alterations**. For each particular human cancer, we collected all corresponding cell lines from the GDSC database. Then, for each of the 16 first-tier metabolic pathways, we segregated these cancer cell lines into two groups: those with and those without alterations in genes involved in a particular metabolic pathway. Finally, we compared the logarithm transformed IC50 values for each of the 251 anticancer drugs between the two groups of cell lines using the Wilcoxon rank sum test. Subsequently, we only returned drugs that had associated IC50 values which differed between cell lines of human cancers with and without alterations of genes involved in a particular metabolic pathway (Supplementary Fig. 9 and Supplementary file 2). Note that these comparisons were only made in the cases were at least four cell lines had alterations of genes involved in a particular metabolic pathway and at least four other cell lines did not have such alterations.

**Survival analysis**. The Kaplan–Meier method was used to estimate overall survival and the duration of disease-free survival between the HM and the LM supertypes of human cancer[36]. To validate our findings concerning the overall survival of the TCGA HM and LM supertypes, we downloaded an independent dataset of overall survival outcomes from the ICGC data portal[37] for individuals afflicted with tumours of types corresponding to those in the TCGA database. Since the ICGC data portal also contains some cancer datasets from the TCGA, we removed these to return a dataset of 3146 patient tumours that are unique to the ICGC. Next, we classified these ICGC patient tumours into the HM or LM supertype categories based on the TCGA classification label provided within the ICGC database. We then compared the overall survival of these HM and LM patients. Also, the Kaplan–Meier method was applied to assess the survival outcomes of oesophageal adenocarcinoma and uterine corpus endometrial carcinoma patients who had tumours with or without alterations to genes involved in the abacavir metabolism pathway.

Further, for each of the 32 cancer types, we individually classified tumours into two categories: those with either higher or lower alteration frequencies to metabolic genes. Here, we used the median alteration rate within each cancer type as the cut-off point for dichotomising the tumours into higher and lower metabolic gene alteration frequency categories. For each of the 32 cancer types, we compared the clinical outcomes of patients by comparing the OS and DFS periods between them (see Supplementary file 2).

The Kaplan–Meier method was also used to estimate OS and DFS between the subgroups of HM tumours that we identified using DBSCAN.

**Statistics and reproducibility**. All statistical analyses were performed in MATLAB 2019a. Fisher's exact test was used to assess associations between categorical variables. The independent sample Student t-test or the Wilcoxon rank sum test and the one-way Analysis of Variance were used to compare continuous variables where appropriate. Statistical tests were considered significant at $p < 0.05$ for single comparisons, whereas the p-values of multiple comparisons were adjusted using the Benjamini–Hochberg method.

**Reporting summary**. Further information on research design is available in the Nature Research Reporting Summary linked to this article.

**Ethical approval**. The University of Cape Town; Health Sciences Research Ethics Committee IRB00001938 approved the protocol of this study. This study involved the analysis of publicly available datasets that were collected by the TCGA, ICGC, GDSC and other databases from consenting participants. All methods were performed following the relevant policies, regulations and guidelines provided by the TCGA, ICGC, GDSC and other databases for analysing their datasets and reporting of the findings.

## Data availability

The data that support the findings of this study are available from the following repositories: cBioPortal (https://www.cbioportal.org/), Genomics of Drug Sensitivity in Cancer (https://www.cancerrxgene.org/), and International Cancer Genome Consortium (https://icgc.org/). The source data in cancer studies and genomic alterations are shown in Supplementary Data 1. Analyses used to show dose-response differences are presented in Supplementary Data 2. Full lists of differentially expressed genes are presented in Supplementary Data 3.

## Code availability

Custom MATLAB code used for data processing and analysis is freely available at: https://github.com/smsinks/Pancancer-Metabolic-Gene-Alterations. The repository includes some predownloaded datasets and conversion files required to reproduce the analysis.

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

## Acknowledgements

Student bursary funding for this project was provided by H3ABioNet, supported by the National Institutes of Health Common Fund under grant number U24HG006941. The content of this publication is solely the responsibility of the authors and does not necessarily represent the official views of the National Institutes of Health.

## Author contributions

The study was conceptualised by M.S., N.M. and D.P.M. The methodology was designed by M.S., N.M. and D.P.M. The formal analysis of the data was performed by M.S. The writing of the original draft manuscript was done by M.S. and D.P.M. The reviewing and editing of the manuscript was done by M.S., N.M. and D.P.M. All visualisation of data was done by M.S.; M.S. was supervised by N.M. and D.P.M.

## Competing interests

The authors declare no competing interests.
