## [Peer Review File · Communications Biology]

Reviewers' comments:

Reviewer #1 (Remarks to the Author):

This manuscript describes the association of metabolic gene "alterations" (mutations plus copy number variations) and clinical outcomes in cancers annotated by TCGA. These alterations also have corresponding metabolic perturbations at the transcriptomic level. Moreover, drug sensitivity may be affected by the frequency of metabolic gene alterations. The most important finding was that a high frequency of alterations in metabolic genes (HM) was associated with more aggressive clinical behavior and different drug sensitivities.

The authors have taken an interesting approach, and I would like to see this work published. This approach is novel, and I believe that the authors have identified an important biological phenomenon. However, I think that major revision will first be required.

The first thing that struck me was the use of the term "alterations". This was confusing and should have been clarified right away. It was not until I read the Methods section that I came to understand that "alterations" meant a sum of mutations and copy number variations (CNVs). A definition should appear in the main body text.

My initial impression was that "alterations" were mutations, and this was reinforced upon review of Figure 1B ("Most Mutated Metabolic Genes"). Figure 1B could be revised to show how frequently each metabolic gene is mutated and how frequently each gene has a CNV.

The premise of the work is based on the dichotomization of tumors according to the frequency of metabolic gene alterations. It will be important to clarify how that dichotomization was applied – how was the cutoff defined? It would also be useful to depict (graphically) what the contribution of mutations and CNVs is. Also, how many tumors were HM and LM in each tumor type?

Abstract: "We further find that there are two metabolically distinct cancer supertypes." I believe that this is an overrepresentation of the findings. I am not convinced with the data provided that there is any metabolic homogeneity in the two supertypes referred to. What distinguishes these two groups is the frequency of mutation and CNV in metabolic genes. The metabolic features of these two variants would be expected to have a wide variety of metabolic features. (They are not metabolically distinct.)

The statement in Line 290 emphasizes this: "Overall, these results indicate that while gene expression profiles are relatively conserved among HM cancers, they are more diverse in the LM cancers."

Figure 2 depicts differences in survival between HM and LM tumors. However, the TCGA and ICGC analyzed diverse tumor types with diverse clinical behaviour and (probably) diverse proportions of HM. Statistically, I would like to see that HM tumor types have a worse survival after correcting for tumor type. Graphically, I would like to see the effects of frequency of alterations in each tumor type (with statistics to illustrate significant differences, correcting for number of tumor types analyzed).

It would be nice to have some Gene Set Enrichment Analyses to understand how HM affects tumor biology, although I do not consider this mandatory for publication. I do think it would add to the value of this work, however.

I am not sure what paragraph 2 on page 12 (Lines 282-288) means.

The introductory paragraphs are well written, but could be more concise.

In the Methods section, the authors should provide details of how the metabolic pathway assignments were made.

Figure 5B: Legend should contain larger colour representations, so it is more clear what colour corresponds to what tumour type.

Figure 6D: could use bigger labels on the X axis.

I think the manuscript would benefit from a more interesting title that captures some of the higher impact features of the work. I think that the title should emphasize that frequency of metabolic gene mutations and copy number variations impacts clinical aggressiveness.

In all, I would encourage publication of this work, but major revisions would greatly improve the presentation.

Oliver Bathe

Reviewer #2 (Remarks to the Author):

Sinkala et al present an articulate and elegant pan-cancer analysis driven by alterations of metabolic pathway genes. Using the TCGA available data, the study classifies 10,528 tumors spanning 32 cancer types into two major subpopulations, HM – high number of alteration, and LM – low number of alterations. This classification and its clinical properties are reproducible in a validation cohort from the International Cancer Genome Consortium (ICGC). The HM group shows worse disease-free survival and overall survival compared to the LM group in a pan-cancer fashion. The LM group shows higher transcriptomic diversity compared to the HM group, indicated by both t-SNE based clustering and straightforward correlation across metabolic genes. Metabolic pathways are grouped in two tiers based on frequency of mutation, one led by lipid, carbohydrate, and amino-acids and derivative metabolism. The second tier is led by glycan metabolism. Finally, the HM and LM groups are shown to associate with potential drug response in a pan-cancer fashion. In corroboration with the difference in transcriptomic diversity, a larger number of drugs are associated with better response in the LM group than in the HM group, based on cell-line/drug response data compiled by the Genomics of Drug Sensitivity in Cancer (GDSC project).

Major Comment:

The study is very promising, but the relative uniformity of the HM group based on the metabolic genes is intriguing. Could the authors use all the available transcriptomic data for the HM group, run t-SNE and a classification method such as DBSCAN or Seurat, and check whether clusters with different overall survival or disease free survival can be found within the HM group? That analysis should give readers a better understanding of the HM tumor group.

We want to thank the reviewers for evaluating our study and providing their insightful and constructive suggestions regarding several aspects of the manuscript. As described below, we have addressed the concerns raised by all the reviewers and made modifications to the manuscript, as suggested.

Please find in this letter a "point-by-point" response to each comment. We have taken all the reviewer's comments seriously and have revised the manuscript accordingly (**red text version**). The comments (*italics*) and the changes made in response (blue) and the **current Page and Line numbers** highlighted in yellow are listed below. References (cyan) made to any **new figures, edited figures and supplementary files**.

Reviewer #2 (Remarks to the Author):

Sinkala et al present an articulate and elegant pan-cancer analysis driven by alterations of metabolic pathway genes. Using the TCGA available data, the studies classifies 10,528 tumors spanning 32 cancer types into two major subpopulations, HM – high number of alterations, and LM – low number of alterations. This classification and its clinical properties are reproducible in a validation cohort from the International Cancer Genome Consortium (ICGC). The HM group shows worse disease-free survival and overall survival compared to the LM group in a pan-cancer fashion. The LM group shows higher transcriptomic diversity compared to the HM group, indicated by both t-SNE based clustering and straightforward correlation across metabolic genes. Metabolic pathways are grouped in two tiers based on frequency of mutation, one led by lipid, carbohydrate, and amino-acids and derivative metabolism. The second tier is led by glycan metabolism. Finally, the HM and LM groups are shown to associate with potential drug response in a pan-cancer fashion. In corroboration with the difference in transcriptomic diversity, a larger number of drugs are associated with better response in the LM group than in the HM group, based on cell-line/drug response data compiled by the Genomics of Drug 115 Sensitivity in Cancer (GDSC project).

Response: We thank the reviewer for praising various aspects of this manuscript and appreciate the constructive comments and suggestions to improve the manuscript further. We have addressed the points raised by the reviewer as below.

Major Comment:

The study is very promising, but the relative uniformity of the HM group based on the metabolic genes is intriguing. Could the authors use all the available transcriptomic data for the HM group, run t-SNE and a classification method such as DBSCAN or Seurat, and check whether clusters with different overall survival or disease-free survival can be found within the HM group? That analysis should give readers a better understanding of the HM tumor group.

Response: We have used all 20,530 of the available transcriptomic datasets of the HM group to run t-SNE and DBSCAN. Here, DBSCAN revealed 8 clusters that displayed statistically significantly differences in durations of overall survival and disease-free survival periods (Page 13: lines 346 – 353, page 28: lines 744 – 746). Please also refer to Figure S7.

Reviewer #1 (Remarks to the Author):

This manuscript describes the association of metabolic gene “alterations” (mutations plus copy number variations) and clinical outcomes in cancers annotated by TCGA. These alterations also have corresponding metabolic perturbations at the transcriptomic level. Moreover, drug sensitivity may be affected by the frequency of metabolic gene alterations. The most important finding was that a high frequency of alterations in metabolic genes (HM) was associated with more aggressive clinical behavior and different drug sensitivities.

Comment 1: *The authors have taken an interesting approach, and I would like to see this work published. This approach is novel, and I believe that the authors have identified an important biological phenomenon. However, I think that major revision will first be required.*

We thank the reviewer for appreciating our study and providing the supportive critiques. We have addressed the suggestions in a pointwise manner as enlisted below.

Comment 2: *The first thing that struck me was the use of the term “alterations”. This was confusing and should have been clarified right away. It was not until I read the Methods section that I came to understand that “alterations” meant a sum of mutations and copy number variations (CNVs). A definition should appear in the main body text.*

Response: We have now put a definition in the summary section and the main body text that “alterations” refers to the sum of mutations and copy number variations (Page 2: lines 42 and 46, Page 5: line 135, Page 6 line 151-152).

Comment 3: *My initial impression was that “alterations” were mutations, and this was reinforced upon review of Figure 1B (“Most Mutated Metabolic Genes”). Figure 1B could be revised to show how frequently each metabolic gene is mutated and how frequently each gene has a CNV.*

Response: We have revised Figure 1B to show both frequently alteration metabolic genes by mutation and CNV.

Comment 4: *The premise of the work is based on the dichotomization of tumors according to the frequency of metabolic gene alterations. It will be important to clarify how that dichotomization was applied – how was the cutoff defined? It would also be useful to depict (graphically) what the contribution of mutations and CNVs is. Also, how many tumors were HM and LM in each tumor type?*

Response: We have now explained in detail how the dichotomization was applied (Page 23: lines 587- 595). Also, we have graphically depicted the contribution of mutations and CNVs for each tumour type and each metabolic pathway (see Figure 1C). Finally, we could state how many tumours were HM and LM for each tumour types because our classification scheme is based on the overall alteration frequencies within each tumour type. However, we have provided a clustering of patient tumours within each cancer type based on the alterations of metabolic genes (Page 26 – 27: lines 647 to 660). Please also refer to Figure S9.

Comment 5: *Abstract: “We further find that there are two metabolically distinct cancer supertypes.” I believe that this is an overrepresentation of the findings. I am not convinced with the data provided that there is any metabolic homogeneity in the two supertypes referred to. What distinguishes these two groups is the frequency of mutation and CNV in metabolic genes. The metabolic features of these two variants would be expected to have a wide variety of metabolic features. (They are not metabolically distinct.)*

Response: we corrected this statement (Page 2: lines 45 – 46).

Comment 6: *The statement in Line 290 emphasizes this: “Overall, these results indicate that while gene expression profiles are relatively conserved among HM cancers, they are more diverse in the LM cancers.”*

Response: thank you for the correction.

Comment 7: *Figure 2 depicts differences in survival between HM and LM tumors. However, the TCGA and ICGC analyzed diverse tumor types with diverse clinical behaviour and (probably) diverse proportions of HM. Statistically, I would like to see that HM tumor types have a worse survival after correcting for tumor type. Graphically, I would like to see the effects of frequency of alterations in each tumor type (with statistics to illustrate significant differences, correcting for number of tumor types analyzed).*

Response: For this analysis, we had only retrieved data for ICGC tumours that were of the same 32 types as those retrieved from the TCGA. Therefore, we have now included a statement in the Methods section to clarify this (Page 27 – 28: lines 696 – 706; Page 8: lines 207 – 209). Also, as requested, we have now evaluated how the frequency of alterations in each tumour type are related to the overall survival and the disease-free survival of patients for each tumour type (Page 9: lines 243 – 247; Page 27 – 28: lines 679 – 689) (please see the supplementary file 2).

Comment 8: *It would be nice to have some Gene Set Enrichment Analyses to understand how HM affects tumor biology, although I do not consider this mandatory for publication. I do think it would add to the value of this work, however.*

Response: We have now performed Gene Set Enrichment Analyses to compare the Gene Ontology Molecular Functions that differ between the HM and LM tumour subtypes (Page 13 – 14: lines 331 – 345 and Pages 26 – 27: lines 632 – 640; please refer to supplementary file 3 and Figure S8).

Comment 9: *I am not sure what paragraph 2 on page 12 (Lines 282-288) means.*

Response: We have rewritten sections of this paragraph to make it easier to comprehend. Briefly, in this paragraph, we compared the correlation of metabolic gene transcripts between all pairs of tumours (Page 12: lines 307 – 317).

Comment 10: *The introductory paragraphs are well written, but could be more concise.*

Response: We have rewritten the introductory paragraphs to make them more concise (Page 3 – 6).

Comment 11: *In the Methods section, the authors should provide details of how the metabolic pathway assignments were made.*

Response: We have explained how we made the metabolic pathway assignments (Page 26: lines 582 – 586).

Comment 12: *Figure 5B: Legend should contain larger colour representations, so it is more clear what colour corresponds to what tumour type.*

Response: We have now made the colour legend markers of bigger in Figure 5.

Comment 13: *Figure 6D: could use bigger labels on the X-axis.*

Response: We have now made the X-axis labels bigger in Figure 6D.

Comment 14: *I think the manuscript would benefit from a more interesting title that captures some of the higher impact features of the work. I think that the title should emphasize that frequency of metabolic gene mutations and copy number variations impacts clinical aggressiveness.*

Response: We have modified the manuscript title to “The Frequency of Metabolic Gene Alterations Impacts the Clinical Aggressiveness and Drug Responses of 32 Human Cancers”.

REVIEWERS' COMMENTS:

Reviewer #1 (Remarks to the Author):

The authors have addressed my concerns. I recommend acceptance.

Reviewer #2 (Remarks to the Author):

The authors have addressed my concerns.